# Ryanodine receptor dispersion disrupts Ca²⁺ release in failing cardiac myocytes

Terje R Kolstad[1,2], Jonas van den Brink[3], Niall MacQuaide[4], Per Kristian Lunde[1], Michael Frisk[1,2], Jan Magnus Aronsen[1,5], Einar S Norden[1,5], Alessandro Cataliotti[1,2], Ivar Sjaastad[1,2], Ole M Sejersted[1], Andrew G Edwards[3], Glenn Terje Lines[3], William E Louch[1,2]*

[1]Institute for Experimental Medical Research, Oslo University Hospital and University of Oslo, Oslo, Norway; [2]KG Jebsen Center for Cardiac Research, University of Oslo, Oslo, Norway; [3]Simula Research Laboratory, Fornebu, Norway; [4]Institute of Cardiovascular Sciences, University of Glasgow, Glasgow, United Kingdom; [5]Bjørknes College, Oslo, Norway

**Abstract** Reduced cardiac contractility during heart failure (HF) is linked to impaired Ca²⁺ release from Ryanodine Receptors (RyRs). We investigated whether this deficit can be traced to nanoscale RyR reorganization. Using super-resolution imaging, we observed dispersion of RyR clusters in cardiomyocytes from post-infarction HF rats, resulting in more numerous, smaller clusters. Functional groupings of RyR clusters which produce Ca²⁺ sparks (Ca²⁺ release units, CRUs) also became less solid. An increased fraction of small CRUs in HF was linked to augmented 'silent' Ca²⁺ leak, not visible as sparks. Larger multi-cluster CRUs common in HF also exhibited low fidelity spark generation. When successfully triggered, sparks in failing cells displayed slow kinetics as Ca²⁺ spread across dispersed CRUs. During the action potential, these slow sparks protracted and desynchronized the overall Ca²⁺ transient. Thus, nanoscale RyR reorganization during HF augments Ca²⁺ leak and slows Ca²⁺ release kinetics, leading to weakened contraction in this disease.
DOI: https://doi.org/10.7554/eLife.39427.001

*For correspondence:
w.e.louch@medisin.uio.no

Competing interests: The authors declare that no competing interests exist.

## Introduction

The basic processes for cardiac excitation-contraction coupling are well described. Depolarization of the sarcolemma triggers the opening of voltage-gated L-Type Ca²⁺ channels (LTCCs), and the resulting Ca²⁺ influx elicits additional Ca²⁺ release via Ryanodine Receptors (RyRs) in the sarcoplasmic reticulum (SR). This process of Ca²⁺-induced Ca²⁺ release leads to a sharp increase in cytosolic Ca²⁺ concentration which initiates cardiomyocyte contraction. In ventricular myocytes, Ca²⁺ release is tightly controlled by the arrangement of LTCCs and RyRs in dyads, with LTCCs present in t-tubules juxtaposed from RyRs across a narrow 12–15 nm dyadic cleft (*Bers, 2001*). The RyRs themselves are organized into clusters; an arrangement that couples their gating, promoting synchronized opening and closing of neighbouring channels (*Marx et al., 2001*; *Sobie et al., 2006*). Recent data have indicated that neighbouring *clusters* of RyRs can also act concertedly if the Ca²⁺ diffusion distance between them is sufficiently short (*Macquaide et al., 2015*). Referred to as 'superclusters' or Ca²⁺ Release Units (CRUs), these functional arrangements of RyR clusters generate Ca²⁺ sparks, the fundamental units of SR Ca²⁺ release in cardiomyocytes (*Cheng et al., 1993*). Ca²⁺ sparks are not only elicited by LTCC opening, but also occur spontaneously during diastole, where spark frequency and geometry can be measured to assess CRU function. While Ca²⁺ sparks are an important source of RyR-mediated Ca²⁺ leak from the SR, 'silent' or 'non-spark' events also occur, and involve the opening of a subset of RyRs within a CRU; so-called 'quarky' release (*Brochet et al., 2011*).

**eLife digest** The muscle cells of the heart coordinate how they contract and relax in order to produce the heartbeat. During heart failure, these cells become less able to contract. As a result the heart becomes inefficient, pumping less blood around the body.

For the cardiac muscle cells to contract, the levels of calcium ions in the cells needs to rapidly increase. In failing hearts, these increases in calcium ion levels are smaller, slower and less well coordinated. It was not known what causes these changes, making it difficult to treat heart failure.

Calcium ions are released in cardiac muscle cells through protein channels called ryanodine receptors. These receptors form clusters that allow them to synchronize when they open and close. Could the reorganization of ryanodine receptors account for the problems seen in failing hearts? To investigate, Kolstad et al. examined rat hearts using a technique called super-resolution microscopy. This showed that the clusters of ryanodine receptors break apart during heart failure to form smaller clusters. Further experiments showed that calcium ions 'leak' from these smaller clusters, reducing the amount of calcium that can be released into cardiac muscle cells during each heartbeat. Released calcium also spreads between the dispersed clusters, resulting in a slower rise of the calcium levels in the cells. Both changes contribute to weakened contractions of cells in failing hearts.

Therefore, heart failure can be traced back to very small rearrangements of the ryanodine receptors. This understanding will help researchers as they investigate new ways to treat heart failure.

DOI: https://doi.org/10.7554/eLife.39427.002

Impaired cardiomyocyte $Ca^{2+}$ homeostasis is believed to importantly contribute to reduced cardiac contractility and arrhythmogenesis in heart failure (HF). SR $Ca^{2+}$ release is reduced and slowed in this condition, and these changes have been linked to altered dyadic structure (*Louch et al., 2010*). We and others have observed marked remodeling of the t-tubular system in failing cardiomyocytes, while RyRs remain predominantly distributed along z-lines (*Song et al., 2006*; *Louch et al., 2006*; *Heinzel et al., 2008*). Thus, the coupling between LTCCs and RyRs is disrupted, with 'orphaned' CRUs exhibiting delayed $Ca^{2+}$ release only after trigger $Ca^{2+}$ diffuses from intact dyads. However, abnormal gaps occurring between t-tubules only account for a fraction of the overall de-synchronization of $Ca^{2+}$ release in HF (*Louch et al., 2006*; *Øyehaug et al., 2013*). This suggests that other alterations might also occur, perhaps at the nanometer scale of CRU organization, which hinder efficient triggering of $Ca^{2+}$ release. CRU reorganization could in principle contribute to increased $Ca^{2+}$ leak, including silent leak, which is a hallmark of heart failure (*Zima et al., 2010*; *Walker et al., 2014*). Exaggerated $Ca^{2+}$ leak in failing cells has been linked to reduced SR $Ca^{2+}$ content and depressed contractile function, elevation of resting $Ca^{2+}$ levels and impaired relaxation, pro-arrhythmic early and delayed afterdepolarizations, and energetic inefficiency as $Ca^{2+}$ is redundantly cycled (*Bers, 2014*). Thus, a detailed understanding of CRU structure and function in failing cells is critical.

The advent of super-resolution microscopy techniques has markedly improved our ability to visualize and quantify CRU organization (*Baddeley et al., 2009*; *Macquaide et al., 2015*; *Jayasinghe et al., 2018*). However, these techniques have not previously been employed to examine RyR configuration in HF. Using direct stochastic optical reconstruction microscopy (dSTORM), we presently report that CRUs become dispersed in failing myocytes, as RyR clusters are broken apart. With the aid of mathematical modeling, we directly link these changes in CRU structure to experimentally measured increases in RyR leak and slowed SR $Ca^{2+}$ release, identifying a novel mechanism underlying pathological remodeling of $Ca^{2+}$ homeostasis in this disease.

## Results

### dSTORM imaging reveals dispersion of CRUs in failing myocytes

Imaging was performed on isolated, fixed cardiomyocytes with antibody labelling of RyR2. Using diffraction-limited confocal imaging (resolution ≈250 nm) and Structured Illumination Microscopy (SIM,

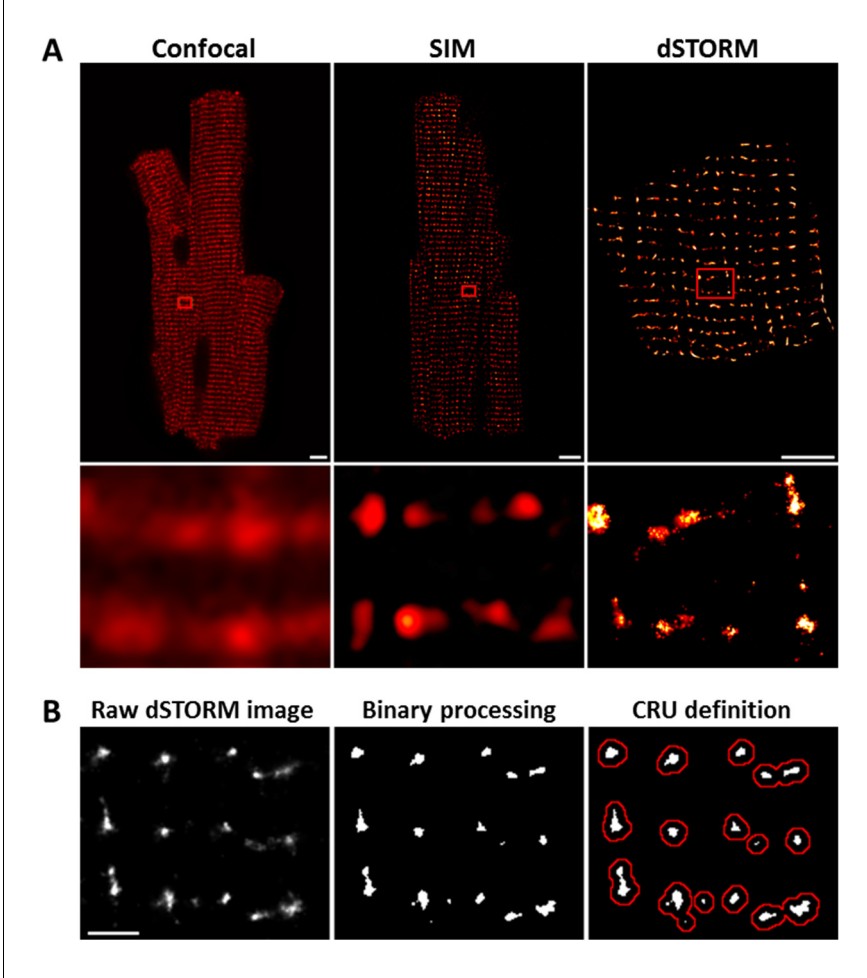

**Figure 1.** dSTORM imaging enables quantification of RyR localization within $Ca^{2+}$ release units (CRUs).  RyR imaging was performed with antibody labelling of isolated and fixed rat ventricular cardiomyocytes. (**A**). Imaging of RyRs with confocal microscopy (left panel) or Structured Illumination Microscopy (SIM, centre panel) revealed a predominantly striated pattern of RyR localization across cells, but individual CRUs could not be discerned (magnified regions in lower panels). dSTORM imaging provided markedly improved spatial resolution enabling identification of RyR clusters (scale bars = 5 µm). (**B**). Quantification of RyR localization was performed by fitting raw images to a 30 × 30 nm grid (*Baddeley et al., 2009*), and performing thresholding to create binary images; an RyR was counted as present if > half the area of a 30 nm square was suprathreshold. CRUs were defined as collections of RyR clusters with an edge-to-edge distance < 150 nm (*Macquaide et al., 2015*) (red boundaries) or < 100 nm (*Baddeley et al., 2009*; *Hou et al., 2015*). (Scale bar = 2 µm).

DOI: https://doi.org/10.7554/eLife.39427.003

resolution ≈120 nm), the localization of RyRs along z-Lines was clearly apparent, but organization of RyRs within CRUs was not discernable (*Figure 1A*). With dSTORM imaging, spatial resolution was markedly improved (mean localization precision = 21 ± 3 nm) enabling detailed CRU geometry to be assessed. For analysis of RyR cluster and CRU configuration, acquired raw images were fitted to a 30 × 30 nm grid, corresponding to the quatrefoil structure of the RyR protein (*Baddeley et al., 2009*). Thresholding was then performed to create binary images (*Figure 1B*), enabling quantification of RyR clusters, with an RyR counted as present if >half the area of a 30 nm square was above threshold. RyR clusters were defined by occupied, neighbouring grid positions, and CRUs were delineated by collecting neighbouring RyR clusters located within 150 nm (*Macquaide et al., 2015*) (red boundaries in *Figure 1B*) or 100 nm (*Baddeley et al., 2009*; *Hou et al., 2015*) (*Figure 2—source data 1*). RyR organization was compared in cardiomyocytes from rats with post-infarction HF and cells from Sham-operated controls. Overall RyR expression was similar in Sham and HF, as evidenced by

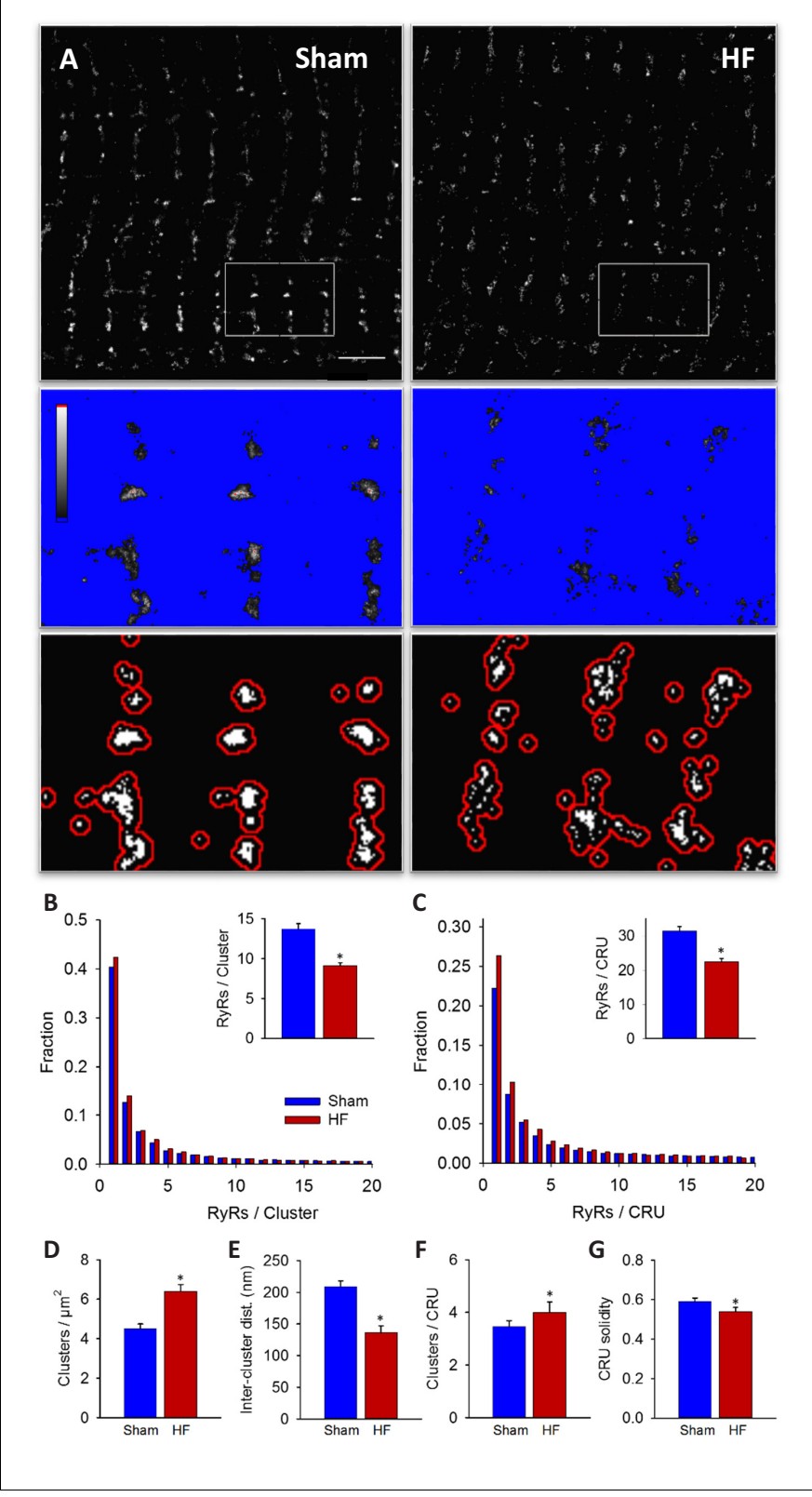

**Figure 2.** RyRs are dispersed in failing cardiomyocytes. Alterations in nanoscale RyR organization were examined in cardiomyocytes from rats with post-infarction heart failure (HF). Representative images show that macroscale organization of RyRs was similar in HF and Sham-operated controls (**A**), upper panels). However, nanoscale examination revealed that RyR clusters were broken apart in HF. For the magnified regions in (**A**), conversion from

*Figure 2 continued on next page*

*Figure 2 continued*

raw dSTORM to binary images is shown in the middle and lower panels (saturation levels indicated by high-low look-up table). Mean measurements showed fewer RyRs per cluster in failing cells, with an increased fraction of small clusters (B). Dispersion of RyR clusters into smaller fragments resulted in an increased overall number of clusters (D), reduced inter-cluster distances (E) and inclusion of more clusters in each CRU (F). Overall CRU composition became less solid in failing cells ((G), assessed by convex-hull analysis), as the average CRU contained fewer RyRs (C). See *Figure 2—source data 1* for analysis of 100 nm vs 150 nm CRU inclusion criterion ($n_{cells}$ = 46, 50 in Sham, HF; *=P < 0.05 vs Sham).

DOI: https://doi.org/10.7554/eLife.39427.004

The following source data and figure supplement are available for figure 2:

**Source data 1.** Quantification of RyR organization using 150 vs 100 nm CRU inclusion criteria.
DOI: https://doi.org/10.7554/eLife.39427.006
**Figure supplement 1.** HF in post-infarction rats is not associated with altered expression of RyR, BIN1, or Junctophilin-2.
DOI: https://doi.org/10.7554/eLife.39427.005

---

Western blotting of ventricular homogenates (*Figure 2— figure supplement 1*), and equivalent RyR labeling density in cardiomyocytes (41.9 ± 1.4 RyR/µm, 40.4 ± 1.3 RyR/µm in Sham, HF respectively). In both groups, RyR staining showed a predominantly transverse, striated pattern (*Figure 2A*). However, despite rather similar organization of RyRs at the macroscale, nanoscale dSTORM imaging revealed fragmentation of RyR clusters in failing cardiomyocytes (see insets in *Figure 2A*). Cluster breakup resulted in a reduction in the number of RyRs per cluster, and a greater proportion of small clusters in HF (*Figure 2B*). The overall number of clusters increased accordingly in failing cells

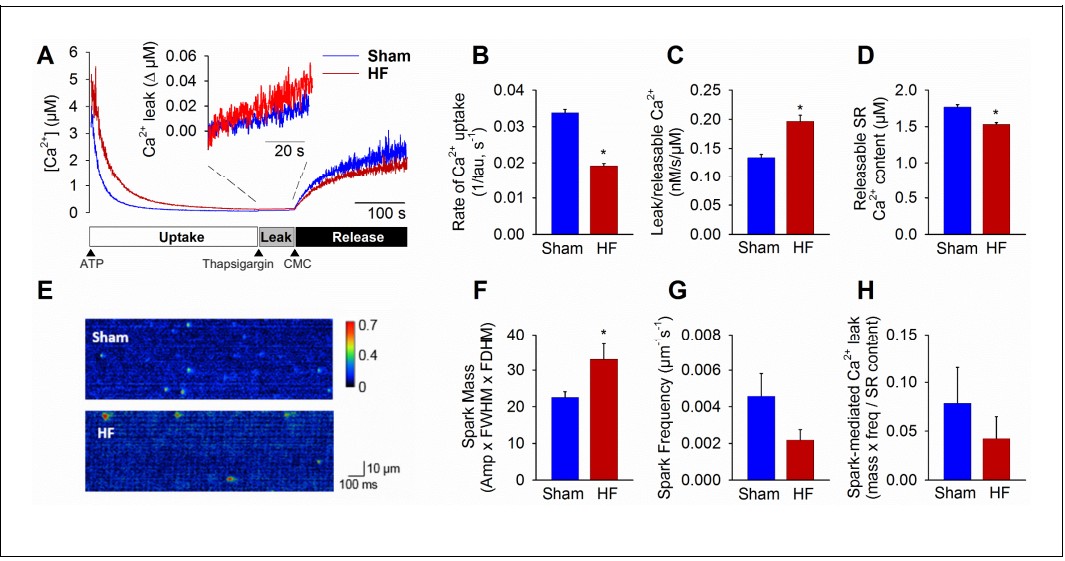

**Figure 3.** Failing cardiomyocytes exhibit increased 'silent' RyR leak. Total RyR-mediated $Ca^{2+}$ leak was assessed in SR microsomes using fura-2 fluorescence (**A**). Vesicular $Ca^{2+}$ uptake was initiated by addition of ATP, and halted by addition of thapsigargin. SR $Ca^{2+}$ leak was estimated as the thapsigargin-induced rate of rise of $[Ca^{2+}]$, normalized to releasable SR content (rise in $[Ca^{2+}]$ induced by the RyR opener 4-chloro-m-cresol, CMC). While the rate of SR $Ca^{2+}$ uptake was reduced in HF relative to Sham (**B**), total RyR leak was increased (**C**) even with a slight reduction in the releasable $Ca^{2+}$ store (**D**). (n = 8 from 3 Sham hearts, 7 from 3 HF hearts; *=P < 0.05 vs Sham). To assess whether elevated SR leak in failing cells could be attributed to $Ca^{2+}$ sparks, line-scan confocal imaging of resting cardiomyocytes was employed (**E**). $Ca^{2+}$ spark mass was increased in HF relative to Sham (**F**), due to augmented spark geometry (*Figure 5B*). However, since spark frequency tended to be reduced (**G**), overall $Ca^{2+}$ spark-mediated leak was similar in Sham and HF (**H**). These results are consistent with increased 'silent' non-spark-mediated SR leak in HF cardiomyocytes. (FWHM = full width at half maximum, FDHM = full duration at half maximum; $n_{cells}$ = 43 in Sham, 50 in HF; *=P < 0.05 vs Sham).
DOI: https://doi.org/10.7554/eLife.39427.007

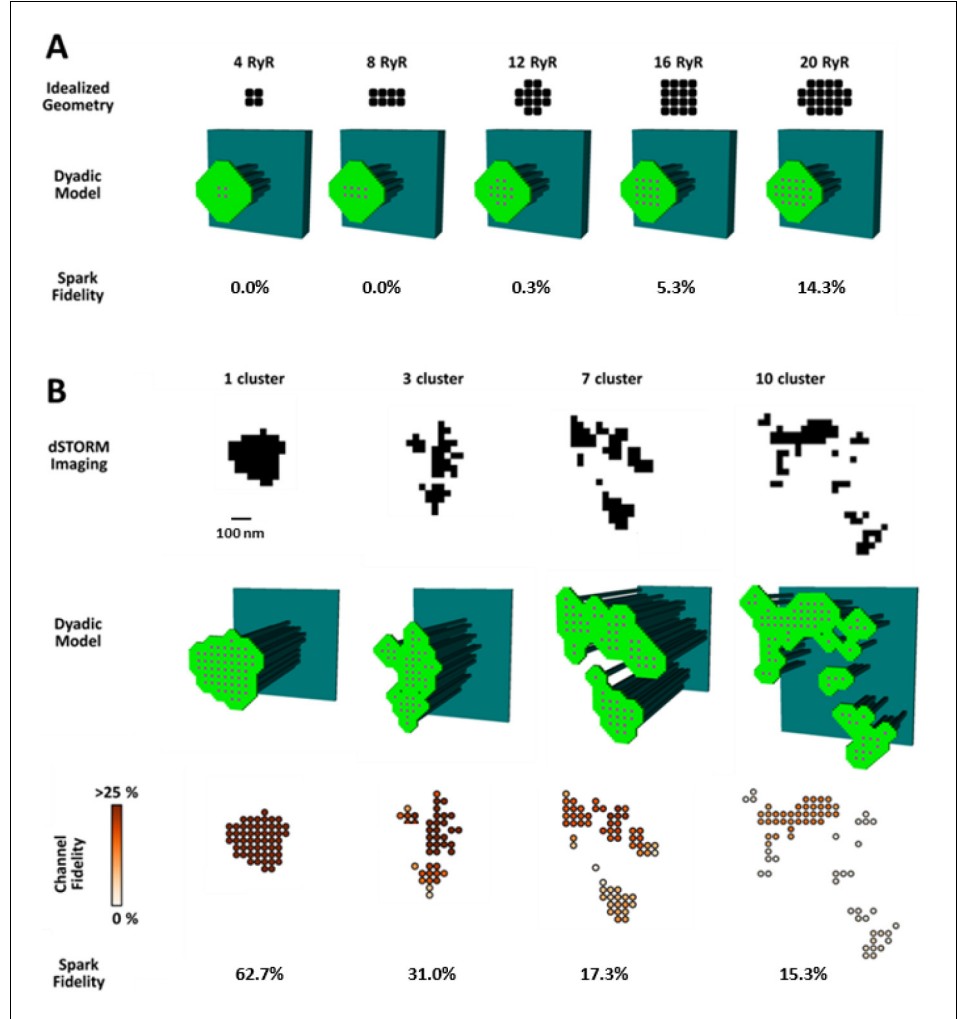

**Figure 4.** CRU dispersion provides the structural basis for silent RyR leak in HF. A mathematical model of the dyad was employed to examine the effects of CRU dispersion on $Ca^{2+}$ sparks and non-spark mediated RyR leak. (**A**). As dSTORM imaging indicated an increased fraction of small CRUs in HF (**Figure 2C**), small idealized CRUs were initially modelled with as few as 4 RyRs. Simulated $Ca^{2+}$ sparks (300 consecutive simulations) were never detected for the smallest CRUs, based on an experimentally determined spark detection threshold of $\Delta F/F_0 = 0.4$. Higher probability of visible spark generation (fidelity) was observed for larger CRUs. (**B**). Real CRU geometries obtained by dSTORM imaging were employed to simulate sparks from larger dyads. Four configurations were modelled with varying numbers of constituent RyR clusters, but similar total RyR number ($\approx 55$). While the single-cluster CRU exhibited high $Ca^{2+}$ spark fidelity, lower probability of spark generation was observed in dispersed, multi-cluster CRUs (fidelity indicated by colour scale). These data support that CRU rearrangement during HF promotes silent RyR leak, due to an increased fraction of both small CRUs as well as larger CRUs with dispersed, irregular configurations.

DOI: https://doi.org/10.7554/eLife.39427.008

The following source data and figure supplement are available for figure 4:

**Source data 1.** Morphological characteristics for the 4 dSTORM-generated geometries.

DOI: https://doi.org/10.7554/eLife.39427.010

**Figure supplement 1.** Development and characterization of the mathematical dyadic model.

DOI: https://doi.org/10.7554/eLife.39427.009

---

(**Figure 2D**), and inter-cluster distance was reduced (**Figure 2E**; see **Figure 2—source data 1** for mean data across animals). Consistent with fragmentation of clusters into smaller adjacent groupings, the number of clusters contained in a CRU increased in HF (**Figure 2F**), although the number of RyRs per CRU decreased (**Figure 2C**) since RyR clusters were markedly reduced in size. Convex hull

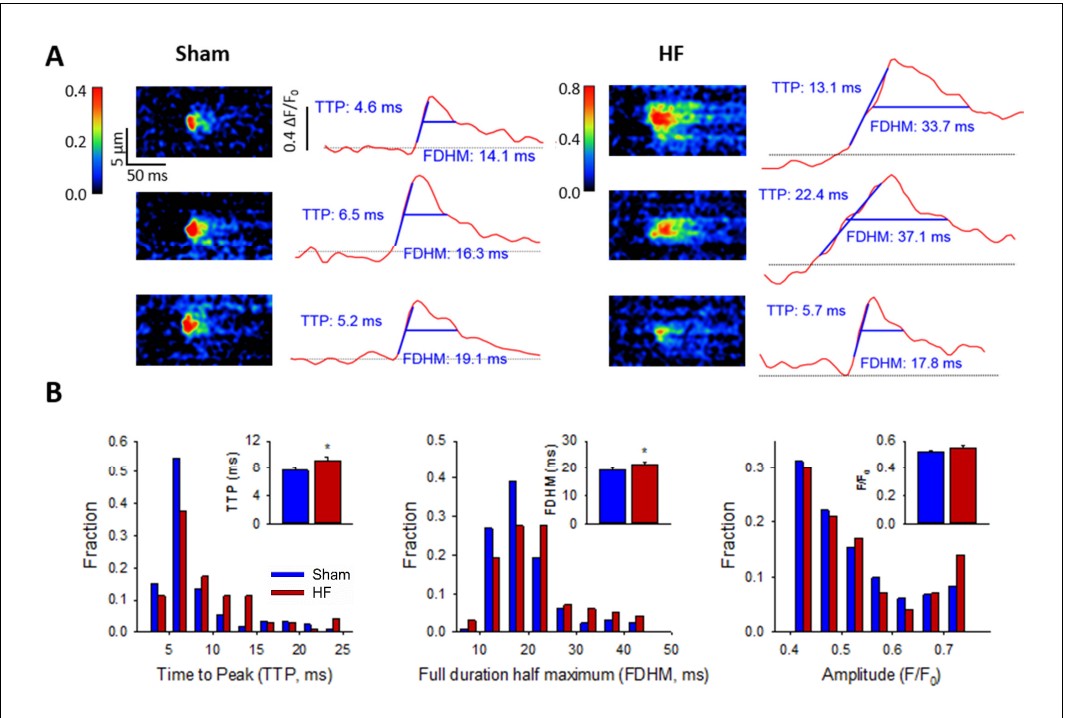

**Figure 5.** Ca$^{2+}$ spark kinetics are slowed in HF. (**A**) Representative line-scan images of Ca$^{2+}$ sparks in Sham and HF, selected from the cell-wide scans presented in *Figure 3E*. Temporal profiles (right panels) show that spark kinetics were generally tightly constrained in Sham, with low values for both time to peak (TTP) and duration (full duration at half maximum, FDHM). Although many sparks were also brief in HF cells, a subset of sparks exhibited slowed kinetics. (**B**) Distributions of measurements for TTP and FDHM were right-shifted in HF, and mean values were significantly increased. Spark magnitudes tended to be larger in HF than Sham. (n$_{sparks}$ = 130, 100 from 75, 72 cells in Sham, HF; *=P < 0.05 vs Sham).

DOI: https://doi.org/10.7554/eLife.39427.011

analysis (see methods) revealed a consequent decrease in CRU solidity in HF (*Figure 2G*). Thus, RyR reorganization in failing cells resulted in CRUs with a more sparse, dispersed configuration of smaller sub-clusters.

## RyR dispersion in HF augments 'silent' RyR Ca$^{2+}$ leak

We examined the functional implications of altered nanoscale organization of RyRs, first hypothesizing that RyR dispersion would augment SR Ca$^{2+}$ leak in failing cardiomyocytes. Total Ca$^{2+}$ leak was assessed in SR microsomes obtained from the left ventricle of Sham and failing hearts.

Following initiation of microsomal Ca$^{2+}$ uptake by addition of ATP, SERCA function was halted by thapsigargin treatment to reveal RyR-mediated Ca$^{2+}$ leak (*Figure 3A*). The Ca$^{2+}$ leak rate was markedly higher in HF compared to Sham (inset in *Figure 3A*, mean data in *Figure 3C*). In agreement with previous work (reviewed in *Bers, 2006*), we additionally observed significantly slowed SR Ca$^{2+}$ uptake in HF (*Figure 3B*) and lower Ca$^{2+}$ content (*Figure 3D*). Ca$^{2+}$ spark-mediated RyR leak was assessed by confocal linescan imaging of freshly-isolated cardiomyocytes (*Figure 3E*). While the average Ca$^{2+}$ release per spark (spark mass) was significantly increased in HF cells, this effect was offset by a tendency toward lower spark frequency (*Figure 3F,G*). Indeed, overall spark-mediated Ca$^{2+}$ leak was similar in HF and Sham cells (*Figure 3H*). Since we observed an increase in total RyR leak in HF, these results are consistent with augmented 'silent', non-spark mediated leak in failing cells.

To investigate whether increased silent Ca$^{2+}$ leak could be linked to RyR dispersion, we employed a mathematical model of the dyad (illustrated schematically in *Figure 4—figure supplement 1A*) that enabled simulation of Ca$^{2+}$ sparks with varied placement of RyRs within the CRU. We first incorporated small idealized CRUs containing as few as 4 RyRs (*Figure 4A*), as our dSTORM imaging

indicated that HF cells contain an increased fraction of small CRUs (*Figure 2C*). During repeated simulations, a single RyR was opened at a random position within the CRU, and subsequent triggered RyR openings were allowed to proceed stochastically. Simulated $Ca^{2+}$ release events with amplitudes $\Delta F/F_0 \geq 0.4$ were defined as sparks, based on the detection threshold determined experimentally (see methods). $Ca^{2+}$ release from the smallest CRUs was never detected, but a progressively greater proportion of events yielded visible sparks as the number of RyRs in these idealized dyad geometries was increased (*Figure 4A*). These results support the assertion that an increased fraction of small CRUs in HF promotes undetectable, silent $Ca^{2+}$ leak.

We next examined whether dispersion of clusters in larger more realistic CRUs could similarly contribute to increased silent $Ca^{2+}$ leak in failing cells. To this end we incorporated real CRU geometries obtained by dSTORM imaging into the model (*Figure 4B*). Four CRUs were selected containing roughly the same number of RyRs, but with different numbers of RyR clusters (1, 3, 7 or 10 clusters). As in the simulations described above for idealized CRU geometries, a single, randomly chosen RyR was opened in each simulation, to determine the likelihood that such triggering would result in a detectable $Ca^{2+}$ spark. While relatively high fidelity spark generation was observed for the single-cluster CRU, $Ca^{2+}$ release was more rarely observed to propagate between clusters, and spark fidelity was significantly lower in multi-cluster CRUs (*Figure 4B*). This reduced efficiency of $Ca^{2+}$ spark triggering in dispersed CRUs partly resulted from greater $Ca^{2+}$ diffusion distance between neighbouring clusters, as demonstrated by progressively increasing the distance between RyR clusters in an idealized dyad (*Figure 4—figure supplement 1B*). Furthermore, released $Ca^{2+}$ is less efficiently confined in the dyadic space when the junctional SR has a more distributed and irregular shape. This latter point was demonstrated in the model by altering the amount of junctional SR surrounding the CRU; increasing junctional SR 'padding' increased spark fidelity in both idealized dyads (*Figure 4—figure supplement 1B*) and dSTORM-based geometries (*Figure 4—figure supplement 1C*).

In summary, these results indicate that nanoscale reorganization of RyRs in HF promotes non-spark-mediated SR $Ca^{2+}$ leak by two mechanisms: (1) by creating smaller CRUs which produce $Ca^{2+}$ release events below the detection limit, and (2) by creating more distributed CRU configurations in which multiple RyR clusters are less likely to co-operatively generate sparks.

## CRU dispersion in HF causes slowing of $Ca^{2+}$ sparks

We next hypothesized that CRU dispersion would slow cardiomyocyte $Ca^{2+}$ release; a hallmark of HF. Representative confocal recordings of $Ca^{2+}$ sparks and their temporal profiles are shown in *Figure 5A*. Spark kinetics in Sham cells generally exhibited rapid rising and declining phases. While some sparks also showed fast kinetics in HF cells, others were markedly slow to rise and decay (*Figure 5A*). Indeed, measurements of spark rise time and duration exhibited broader distributions and were, on the average, prolonged in HF compared to Sham (*Figure 5B*). To investigate whether CRU dispersion in HF could underlie slowing of $Ca^{2+}$ spark kinetics, we again employed our mathematical model with dSTORM-based CRU configurations. During the simulations, the time to opening of each RyR was registered, and the time course of the overall $Ca^{2+}$ spark determined. Representative simulations show that RyR opening times were delayed in the dispersed, multi-cluster CRUs compared to the solid, single-cluster CRU (*Figure 6A*). Simulations of $Ca^{2+}$ spark time courses further showed that the delayed RyR openings in multi-cluster CRUs resulted in more variable kinetics and overall slowing of spark rise time (*Figure 6B*, mean data *Figure 6C*), reproducing experimental observations. Of note, although CRUs were observed to contain fewer RyRs in HF than Sham (*Figure 2C*), simply reducing the RyR number to an equivalent degree in the mathematical model did not markedly alter $Ca^{2+}$ spark kinetics (*Figure 6—source data 1*), further confirming a key role of CRU fragmentation in failing cells.

## Slow $Ca^{2+}$ sparks promote slowing and de-synchronization of the $Ca^{2+}$ transient

Finally, we examined the consequences of increased variability in $Ca^{2+}$ spark kinetics for the $Ca^{2+}$ transient in failing cells. We observed that field-stimulated $Ca^{2+}$ transients were significantly slower to rise in HF than Sham (*Figure 7A–C*). This slowing of $Ca^{2+}$ release was associated with marked de-synchronization of the $Ca^{2+}$ transient, which we quantified by measuring the variability in time to reach half-maximal fluorescence ($TTF_{50}$) across the cell (see lower panels in *Figure 7A*). This

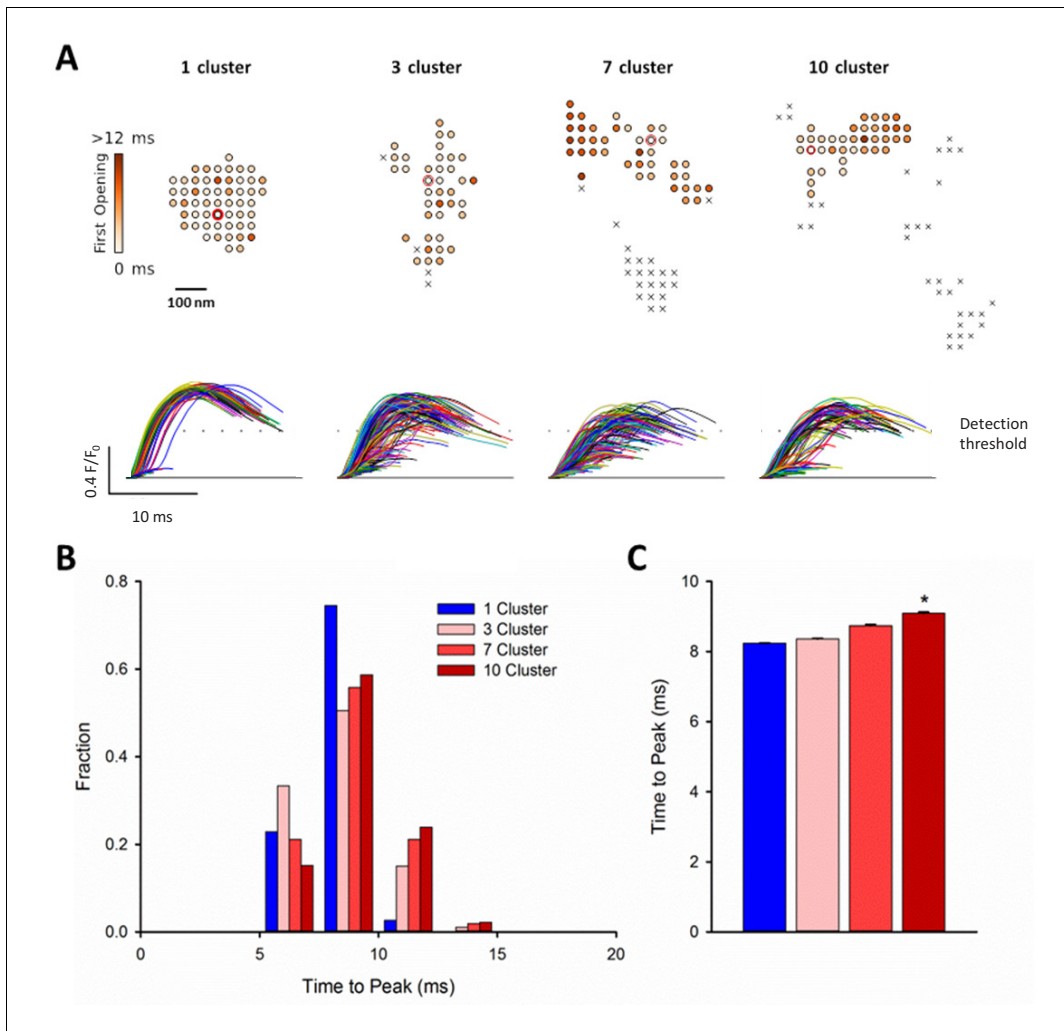

**Figure 6.** RyR dispersion during HF results in slowing of $Ca^{2+}$ sparks. To examine whether altered CRU morphology could slow $Ca^{2+}$ spark kinetics in HF, spark profiles were simulated for a variety of dSTORM-derived RyR configurations. (**A**) Sparks were triggered by opening a single RyR (circled) which was randomly placed in consecutive simulations (example RyR opening trajectories are shown in the upper panels, with a family of spark time-courses illustrated below). Time to opening was registered for each RyR in the CRU, and the resultant time course of the $Ca^{2+}$ spark was plotted until the final RyR closure, at which point the simulation was stopped for computational efficiency. Opening times were similar for individual RyRs within a solid, single cluster CRU, and the overall temporal profile of elicited sparks showed rapid kinetics which were rather consistent between consecutive simulations. By contrast, delayed and variable opening times were observed for individual RyRs in multi-cluster CRUs. This resulted in variable and slowed $Ca^{2+}$ spark kinetics with these CRU configurations, as indicated by temporal spark profiles (**A**), a right-shifted distribution of time-to-peak measurements (**B**) and mean data (**C**). (\*=P < 0.05 vs single-cluster CRU).

DOI: https://doi.org/10.7554/eLife.39427.012

The following source data and figure supplement are available for figure 6:

**Source data 1.** CRU size has little effect on $Ca^{2+}$ spark characteristics in the reported range.
DOI: https://doi.org/10.7554/eLife.39427.014

**Figure supplement 1.** Effect of focal plane on $Ca^{2+}$ spark detection and kinetics.
DOI: https://doi.org/10.7554/eLife.39427.013

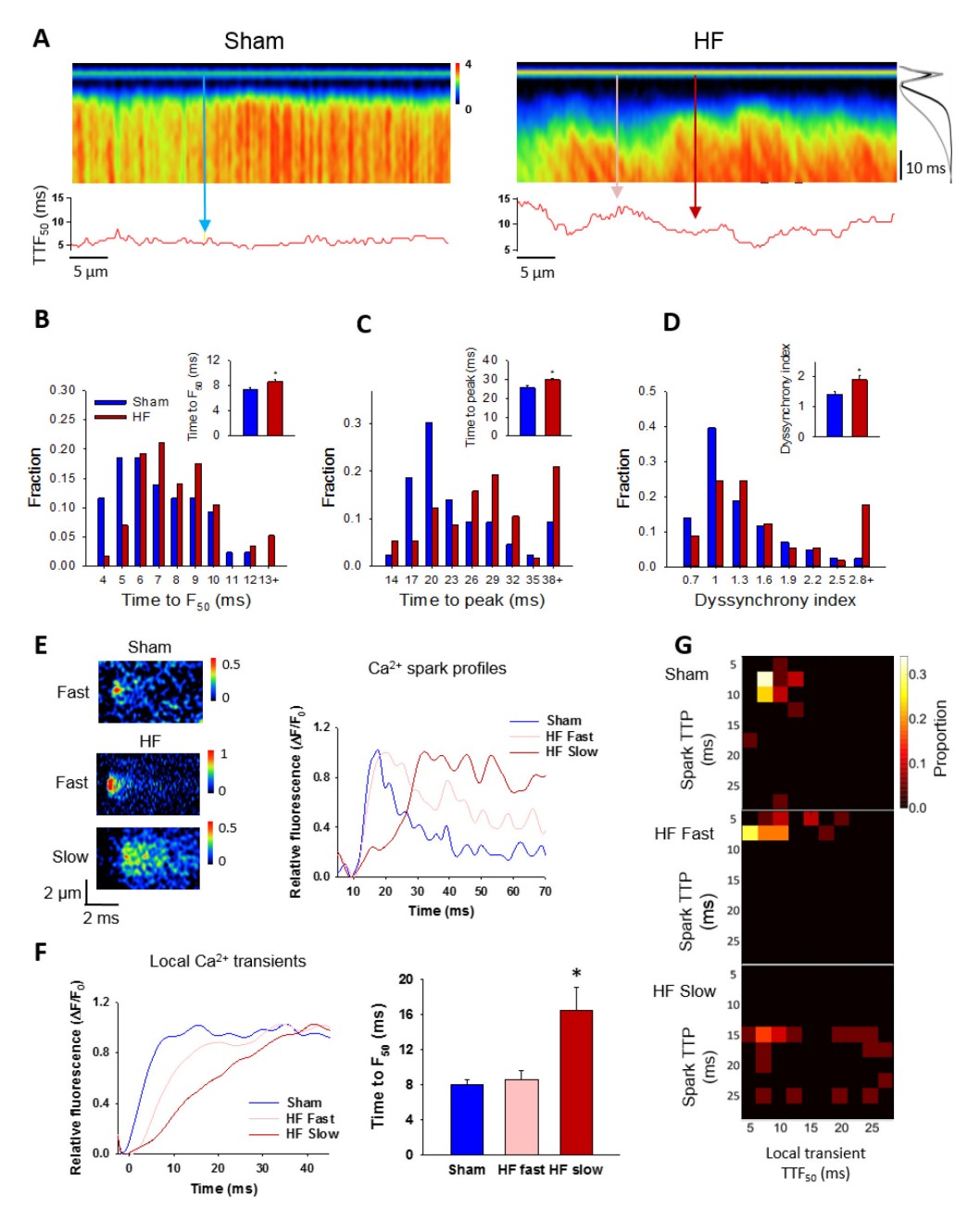

**Figure 7.** Slow Ca²⁺ sparks promote slow, desynchronized Ca²⁺ transients in HF. (**A**) Representative confocal linescan images of Ca²⁺ transients in field-stimulated cells (stimulus illustrated as a horizontal line). The overall Ca²⁺ transient was slowed in HF compared to Sham, as indicated by plots of spatially-averaged Ca²⁺ transients ((**A**), right panel), and measurements of half rise time (TTF₅₀, (**B**)) and time to peak (**C**). Slowed Ca²⁺ transient kinetics included de-synchronization of Ca²⁺ release across HF cells, as indicated by profiles of local TTF₅₀ (lower panels in **A**). The standard deviation of these values, defined as the *dyssynchrony index* (***Louch et al., 2006***), showed a right-shifted distribution in HF compared to Sham (**D**). To examine the relationship between slowed Ca²⁺ spark kinetics and de-synchronized Ca²⁺ transients in HF, local Ca²⁺ transients were examined within 2 µm regions of the linescan centered at the location of recorded sparks. Paired representative recordings of sparks and local Ca²⁺ transients are shown in (**E** and **F**), respectively, corresponding to indicated positions in **A**) (vertical arrows). Local Ca²⁺ release at 'slow' spark locations (rise time > 13 ms) was protracted during the action potential, in comparison with local transients with 'fast' sparks in both HF and Sham (**F**). This association is demonstrated by clustering of locations with slow Ca²⁺ spark and local transient kinetics in 'heat maps' (**G**), and links slowing of

*Figure 7 continued*

$Ca^{2+}$ release kinetics at the level of the single CRU and whole cell. ($Ca^{2+}$ transients: $n_{cells}$ = 43 in Sham, 57 in HF; $n_{fast\ sparks}$= 18 in Sham, 19 in HF; $n_{slow\ sparks}$= 18 in HF; *=P < 0.05 vs Sham).

DOI: https://doi.org/10.7554/eLife.39427.015

The following figure supplement is available for figure 7:

**Figure supplement 1.** T-tubule structure is disrupted during HF.

DOI: https://doi.org/10.7554/eLife.39427.016

'dyssynchrony index' (*Louch et al., 2006*) was significantly increased in HF compared to Sham, with a strongly right-shifted distribution of values (*Figure 7D*). T-tubule disruption in failing cells (*Figure 7—figure supplement 1*) has been previously established in this model of HF (*Frisk et al., 2016*), and is a recognized cause of $Ca^{2+}$ release dyssynchrony in this disease (*Song et al., 2006*; *Louch et al., 2006*; *Heinzel et al., 2008*). We examined whether alterations in $Ca^{2+}$ spark kinetics also promote dyssynchrony, by examining local $Ca^{2+}$ transients within narrow, 2 μm regions of the line scan. These regions were centered at the locations of spontaneous $Ca^{2+}$ sparks observed when electrical pacing was halted. We specifically distinguished between locations with 'slow' sparks, defined by a rise time >13 ms (ie. 1 S.D. > mean rise time in Sham), and remaining 'fast' sparks. By this definition, 24% of sparks in HF cells were defined as slow, while only 13% of Sham sparks fit this definition. Representative examples of such sparks and their temporal profiles are shown in *Figure 7E*, with corresponding positions along the line scan indicated in *Figure 7A*. Local transients from slow spark locations in HF exhibited markedly slower rise times than those from fast spark locations in both HF and Sham (*Figure 7F*). The association between slow sparks and slow local transients was also apparent in 'heat map' plots (*Figure 7G*). These results show that by protracting $Ca^{2+}$ sparks, CRU dispersion during HF slows and desynchronizes the overall $Ca^{2+}$ transient.

## Discussion

In the present study, we have employed dSTORM imaging to reveal key changes in CRU morphology during heart failure. We specifically observed marked dispersion of RyRs, which resulted in a shift towards smaller RyR clusters and CRUs. Remaining larger CRUs became less solid, with more fragmented configurations. Experiments and mathematical modeling linked these changes in RyR arrangement to two central aspects of impaired $Ca^{2+}$ homeostasis in failing cells: increased 'silent' RyR leak and slowing of $Ca^{2+}$ release, which are believed causative for reduced contractility in this condition.

In contrast to lower resolution optical imaging techniques such as confocal and SIM microscopy, the dSTORM technique allows quantification of RyR organization within CRUs. We presently employed this technique with a grid-based quantification method for RyR counting, as previously developed by the Soeller group (*Baddeley et al., 2009*; *Hou et al., 2015*). Using images collected close to the cell surface (depth of 200 – 500 nm, see also Macquaide *et al.* [*Macquaide et al., 2015*]), we calculated that an average cluster contains approximately 14 RyRs in our healthy, Sham-operated rat ventricular cells. This estimation is in close agreement with previous estimates made at the cell surface (14 RyRs/cluster) (*Baddeley et al., 2009*), but considerably lower than estimates made deep within the cell interior (*Hou et al., 2015*), a discrepancy which may reflect regional differences in RyR organization across the cell. However, it should be noted that the grid-based method for RyR counting assumes that the channels lie parallel to the field of view, with a uniform, grid-like configuration; presumptions which are likely less valid when RyRs are visualized internally than at the cell surface. Due to superimposition of internal RyRs along the z-axis, it is therefore likely that present RyR cluster sizes are somewhat underestimated. Assuming that RyR clusters located within 150 nm cooperatively form a CRU (*Macquaide et al., 2015*), we calculated that average CRUs contain roughly 3 – 4 clusters, and a total of ≈30 RyRs. Using a narrower CRU definition, with inclusion of neighbouring clusters within 100 nm (*Baddeley et al., 2009*; *Hou et al., 2015*), reduced average CRU size to ≈25 RyRs (*Figure 2—source data 1*). Despite possible underestimation of RyR numbers/CRU due to methodological issues noted above, these estimates are in relatively close agreement with an electron microscopy tomography study of mouse ventricular cardiomyocytes (*Hayashi et al., 2009*). The authors reported that while there was great variability in CRU size and

geometry, dyadic volume was an order of magnitude lower than previous estimates (*Franzini-Armstrong et al., 1999*; *Scriven et al., 2013*); an average-sized dyad could only hold up to 43 RyRs, with RyRs occupying ≈78% of the dyadic space (≈34 RyRs/CRU). It is estimated that typical sparks result from the activation of somewhat fewer RyRs, ≈20–30 (*Shkryl et al., 2012*). One possible explanation for this discrepancy is that not all RyRs within a CRU may contribute to every spark. Alternatively, previous work based on estimation of the dyadic size available for RyRs (*Hayashi et al., 2009*) may have overestimated RyR number if RyRs are not densely packed (*Asghari et al., 2014*). Our own present estimates have assumed that RyRs located within a fixed distance share the same junctional SR (jSR), which may also be an overestimation since true jSR geometry is unknown. Indeed, our modeling results showed that reducing the degree of jSR 'padding' around each RyR cluster markedly reduces the ability of neighbouring clusters to function as cooperative CRUs (*Figure 4—figure supplement 1B,C*). Future work may address this important issue by simultaneously assessing jSR and RyR arrangement by multi-colour super-resolution imaging, or direct visualization of RyRs and jSR using electron microscopy.

Previous work by Zima *et al.* has shown that silent, non-spark mediated leak is a significant contributor to overall RyR leak (*Zima et al., 2010*). We presently show that silent leak can be partly traced to small CRUs, which produce $Ca^{2+}$ release events that are not detectable experimentally (see also *Walker et al., 2014*). Consistent with previous work (*Baddeley et al., 2009*; *Hayashi et al., 2009*; *Hou et al., 2015*), we observed that many CRUs have very small geometry even in healthy cells; 42% of CRUs contained five or fewer RyRs in Sham (*Figure 2C*). However, larger CRUs can also contribute to non-spark mediated RyR leak, when $Ca^{2+}$ release from CRU sub-clusters does not propagate to remaining clusters; so-called 'quarky' $Ca^{2+}$ release (*Brochet et al., 2011*). Our results show that the decreased fidelity of spark triggering in these dispersed CRUs results from the spacing between neighbouring clusters, which inhibits cooperative, diffusion-based triggering. Furthermore, released $Ca^{2+}$ more easily escapes from less densely packed CRUs, making it less likely to trigger additional RyRs. This finding is in agreement with previous modeling studies showing that spark fidelity declined when clusters deviated from circular and compact shapes (*Walker et al., 2014*), and that RyR activation during triggered release is less likely when CRUs are broken into sub-clusters (*Cannell and Soeller, 1997*). We presently link augmented silent leak in HF to an increased fraction of both small CRUs and CRUs with larger, distributed configurations. These results have important implications. Increased RyR leak during heart failure is widely believed to promote arrhythmia via generation of both early and delayed afterdepolarizations. Furthermore, greater leak also promotes depression of contraction, via loss of SR $Ca^{2+}$ content, and poor relaxation, due to elevation of resting $Ca^{2+}$ (reviewed in *Bers, 2014*). The present results provide a structural basis for these maladaptive functional alterations, and suggest that the nanometer scale of these changes prevented their previous detection by lower resolution imaging techniques.

Slowing of $Ca^{2+}$ release is another key component of the failing phenotype. Prolonged rise time of the $Ca^{2+}$ transient during HF (*Figure 7A–C*) has been previously linked to slowed isolated cardiomyocyte contraction in both animals (*Bokenes et al., 2008*; *Mørk et al., 2009*) and patients (*Davies et al., 1995*). In vivo systolic tissue velocity is also reduced in HF patients (*Vinereanu et al., 2005*) which decreases contractile power. Thus, understanding the mechanisms underlying slowed $Ca^{2+}$ release in failing cells is critical. Mathematical modeling studies showed that while out-of-focus release events can theoretically reduce spark kinetics artefactually, these events rapidly become undetectable as they are shifted further from the focal plane (*Figure 6—figure supplement 1*). Furthermore, there is no clear basis for expecting that dispersion of RyR clusters in HF should change the proportion of out-of-focus sparks relative to control, and for this reason we do not expect that variation in the focal plane has systematically impacted our measured rise times. Our modeling results also suggest that an altered number of RyRs/CRU, due either to methodological under-estimation or loss of RyRs from CRUs in HF, will not in and of itself promote slowing of sparks kinetics (*Figure 6—source data 1*). Indeed, we observed that spark kinetics were relatively insensitive to changes in RyR number for medium-sized CRUs, in agreement with previous work (*Cannell et al., 2013*). Rather, our results point to an important role of CRU dispersion in slowing $Ca^{2+}$ sparks, as multi-cluster dyads exhibited progressive triggering of individual clusters by diffusion of released $Ca^{2+}$ (*Figure 6*). Importantly, sites with slow spontaneous $Ca^{2+}$ sparks were observed to also have slow local triggered $Ca^{2+}$ release during the action potential (*Figure 7*). Thus, dispersed CRUs promote overall slowing and de-synchronization of the $Ca^{2+}$ transient. While these results link visible

spark-mediated leak to slowing of overall Ca$^{2+}$ release, small, undetectable RyR openings (silent leak) might also play an important role. Previous work has shown that RyR clustering allows the channels to be functionally coupled, whereby they exhibit coordinated opening and closing (*Marx et al., 2001*; *Wang et al., 2004*; *Sobie et al., 2006*). Wang and colleagues reported that this thermodynamic stability is lost when RyRs are present alone or with a small number of neighbouring channels, and slow Ca$^{2+}$ release kinetics result (*Wang et al., 2004*). An increased fraction of small CRUs in failing cells may therefore augment slow, but undetectable Ca$^{2+}$ release events which nevertheless contribute to an overall Ca$^{2+}$ transient which is slow and de-synchronized.

Dyssynchronous Ca$^{2+}$ release during heart failure has been previously linked to t-tubule disruption in a large number of studies (reviewed in *Louch et al., 2010*), and we have similarly observed t-tubule disorganization in this post-infarction rat model (*Figure 7—figure supplements 1* and *Frisk et al., 2016*). An important question is therefore whether CRU morphology and slow Ca$^{2+}$ sparks occur independently from t-tubule reorganization, or whether these two aspects of structural remodeling are related. In previous work, we (*Louch et al., 2013*) and others (*Meethal et al., 2007*) employed simultaneous imaging of t-tubules and Ca$^{2+}$ and observed that sparks occurred almost exclusively at t-tubule sites in both healthy and failing cells. Similarly, de-tubulation experiments have been shown to dramatically reduce the occurrence of Ca$^{2+}$ sparks in the cell interior, suggesting that SR-t-tubule junctions are important for spark initiation (*Brette et al., 2005*). It may be postulated, therefore, that CRU dispersion resulting in slow sparks occurs at intact dyads in HF (ie. not at sites with 'orphaned RyRs'), and that t-tubule and CRU remodeling may occur independently. Verification of this point will likely require simultaneous nanoscale imaging of t-tubules and RyRs, as small degrees of t-tubule drift out of dyads may be critical, and not detected by lower resolution imaging techniques. Regardless, we believe that t-tubule and CRU disruption have additive, detrimental effects, resulting in markedly de-synchronized and slowed SR Ca$^{2+}$ release.

What signals RyR dispersion in HF? Emerging data indicate that RyRs are not firmly anchored within the CRU, but exhibit a highly malleable organization dependent on factors such as phosphorylation status and cytosolic Mg$^{2+}$ levels (*Asghari et al., 2014*). However, while changes in such conditions were shown to influence whether RyRs are positioned in a 'checkerboard' or side-by-side arrangement, it is unclear whether they can lead to reorganization of clusters and CRUs on the scale of changes presently observed in HF. Another important dyadic regulator is Junctophilin-2 (JPH-2), which anchors the t-tubule to the SR (*Takeshima et al., 2000*; *Minamisawa et al., 2004*), and interacts with the RyR (*Beavers et al., 2013*; *Munro et al., 2016*). Munro *et al* recently reported that JPH-2 levels regulate RyR clustering; however, while they observed that JPH-2 overexpression triggered the formation of larger RyR clusters, JPH-2 knockdown did not reduce cluster size (*Munro et al., 2016*). Although others have observed JPH-2 loss during HF (*Minamisawa et al., 2004*; *Wei et al., 2010*), we did not presently observe reduced JP-2 protein levels in our rat HF model (*Figure 2—figure supplement 1*), suggesting that JP-2 downregulation is not a prerequisite for reorganization of CRUs (*Figure 2*) or t-tubules (*Figure 7—figure supplement 1*). Another dyadic regulator, BIN1, is a well-established regulator of t-tubule growth and structure (*Hong et al., 2014*), and recent data have suggested that this protein may also attract phosphorylated RyRs to the dyad (*Fu et al., 2016*). Although BIN1 loss has been reported in other HF models (*Lyon et al., 2012*; *Caldwell et al., 2014*), our data indicate that BIN1 expression is unaltered in our rat model (*Figure 2—figure supplement 1*). Thus, we do not believe that BIN1 changes are related to the CRU reorganization presently observed in failing myocytes, and the precise trigger for such changes remains unclear.

In conclusion, our results contribute to an emerging understanding that cardiomyocyte dyads are highly plastic structures. While previous work has shown that t-tubule structure is impressively malleable, and degraded during heart failure, our present findings show that there is also detrimental reorganization of RyRs in this disease. Dispersion of RyRs within the CRU was linked to increased silent RyR leak, slowing of Ca$^{2+}$ sparks, and de-synchronization of the overall Ca$^{2+}$ transient, indicating a novel mechanism underlying impaired contractility in HF.

## Materials and methods

**Key resources table**

| Reagent type (species) or resource | Designation | Source or reference | Identifiers | Additional information |
|---|---|---|---|---|
| Strain, strain background (*Rattus norvegicus*, M) | Male Wistar-Hannover rats | Janvier-labs | RjHan: WI; RGD: 13792727 | |
| Antibody | Mouse-anti-RyR2 primary antibody | ThermoFischer Scientific | Cat# MA3-916; RRID:AB_2183054 | IHC, 1:100; WB, 1:1000 |
| Antibody | Alexa Fluo 647 conjugated goat-anti-mouse secondary Ab | Molecular Probes/Invitrogen | Cat# A-21237; RRID:AB_2535806 | IHC, 1:200 |
| Antibody | Anti-goat IgG-HRP linked whole antibody | R and D Systems | Cat# HAF109; RRID:AB_357236 | WB, 1:3000 |
| Antibody | Mouse IgG HRP linked Whole Ab | GE Healthcare | Cat# NA931V; RRID:AB_772210 | WB, 1:3000 |
| Antibody | Rabbit IgG HRP linked Whole Ab | GE Healthcare | Cat# NA934V; RRID: AB_772206 | WB, 1:3000 |
| Antibody | Goat Anti-GAPDH Polyclonal antibody | Santa Cruz Biotechnology | Cat# sc-20357; RRID:AB_641107 | WB, 1:500 |
| Antibody | Goat Anti-Junctophilin-2 Polyclonal Antibody | Santa Cruz Biotechnology | Cat# sc-51313; RRID:AB_2296391 | WB, 1:1000 |
| Antibody | BIN1 (Amphiphysin II (2F11) Antibody) | Santa Cruz Biotechnology | Cat# sc-23918; RRID:AB_667901 | WB, 1:500 |
| Software, algorithm | Dense Stochastic Sampling Imaging (DSSI) algorithm | SoftWoRx, GE Healthcare | | |
| Software, algorithm | dSTORM image post-processing algorithm | PMID: 26490742 | | Described in the *Github* repository (*Kolstad, 2018*; copy archived at https://github.com/elifesciences-publications/Ryanodine_Receptor_Dispersion_during_Heart_Failure) |
| Software, algorithm | Mathematical Model | PMID: 22495592; PMID: 23708355; this paper | | Described in the *Github* repository (*Kolstad, 2018*; copy archived at https://github.com/elifesciences-publications/Ryanodine_Receptor_Dispersion_during_Heart_Failure) |
| Software, algorithm | SigmaPlot | SigmaPlot | RRID:SCR_003210 | |

## Rat model of post-myocardial infarction congestive HF

All experiments were approved by the Norwegian National Animal Research Authority (project license no. FOTS 5982, 7786), and were performed in accordance with the National Institute of Health guidelines (NIH publication No. 85 – 23, revised 2011) and European Directive 2010/63/EU. Large anterolateral myocardial infarctions were induced in ~300 g male Wistar-Hannover rats, by ligation of the left coronary artery as previously described (*Lunde et al., 2012*). Development of HF was verified six weeks later using a Vevo 2100 echocardiography imaging system (VisualSonics, Toronto, Canada). Inclusion of failing animals was based on established criteria (*Sjaastad et al., 2000*), including dilation of the left atrium (diameter >5 mm) and ventricle, and increased lung weight (>2.5 g). Sham-operated rats served as controls. Experiments were performed over a two year period, using animals from 10 rounds of animal surgery. Sample sizes were determined by power analysis,

assuming that only 50% of post-infarction animals would be included in the final data set, and based on a pilot project of variability in CRU morphology in healthy controls.

## Cell isolation

Cardiac myocytes from failing and Sham-operated rats were isolated using a standard enzymatic dispersion technique (*Louch et al., 2011*). Excised hearts were mounted on a Langendorff setup, and retrogradely perfused through the aorta with $Ca^{2+}$-free solution containing (in mmol/L): 130 NaCl, 25 Hepes, 5.4 KCl, 0.5 $MgCl_2$, 0.4 $NaH_2PO_4$, 5.5 D-glucose, pH 7.4. Once cleared of blood, hearts were then perfused with the above solution including collagenase (2 mg/mL, Worthington Biochemical Corp., Lakewood, NJ, USA) and low $[Ca^{2+}]$ (0.05 mmol/L). After 10 min of digestion, hearts were cut down, minced, and filtered, and isolated cardiomyocytes were allowed to sediment.

## Immunostaining

Isolated cardiomyocytes were transferred to cell culture medium (DMEM 1X, Life Technologies with 10% FBS, Biowest Nuaillé, France and 1% Penicillin-Streptomycin, Sigma), and plated on laminin-coated, glass bottom culture dishes (MatTek corporation, Ashland MA). Staining was performed according to a described protocol (*Swift et al., 2007*), with consecutive steps for chemical fixation (4% Formaldehyde in 1 mol/L HEPES buffer, 10 min), quenching (PBS + 100 mmol/L Glycine, 10 min), permeabilization (PBS + 0,03% Triton X-100, 10 min), and blocking (NaCl 150 mmol/L, $Na_3$ citrate 17.5 mmol/L, 5% goat serum, 3% BSA, 0.02% $NaN_3$, 2 hr). PBS washing was performed in between each step. The cells were then incubated overnight with 1/100 diluted mouse-anti-RyR2 primary antibody (ThermoFischer Scientific, MA3–916) in low blocking buffer, containing 150 mmol/L NaCl, 17.5 mmol/L $Na_3$ citrate, 2% goat serum, 1% BSA, and 0.02% $NaN_3$ at 4°C. This protocol has previously been reported to result in the binding of multiple primary antibodies to each RyR tetramer (*Baddeley et al., 2009*). The following day, cells were washed with PBS and incubated with 1/200 diluted secondary antibody (Alexa Fluo 647 conjugated goat-anti-mouse secondary Ab, Molecular Probes/Invitrogen) in low blocking buffer for 2 hr. Cells were then washed and stored in PBS until image acquisition. Of note, the fab-fragment secondary antibody employed places the fluorescent label far closer to the epitope than traditional antibodies. Thus, under our experimental conditions, the steric error is generally <10 nm, and dwarfed by the localization of the dSTORM technique ($\approx$20 nm, see below). dSTORM imaging was performed using an OMX V4 system (Applied Precision, GE Healthcare) with a 60 × 1.49 NA TIRF objective (Olympus), a pco.edge sCMOS camera (PCO), a 100 mW 642 nm laser, and a 683/40 emission filter. Focusing was performed with a 30V300nanoX CL focusing unit (Piezosystem, Jena). Cells were placed in 'switching buffer' (0.5 mg/mL glucose oxidase, 40 μg/mL catalase, 10% wt/vol glucose, 50 mmol/L β-mercaptoethylamine in Tris-buffer, pH 8.0, all Sigma–Aldrich), and fluorophores were pushed into the dark state by illumination with the 642 nm laser at a highly inclined, but sub-TIRF angle (Highly Inclined and Laminated Optical sheet, HILO; *Tokunaga et al., 2008*). Spontaneous blinking occurred without the use of an activation laser, and was recorded at a depth of 200–500 nm during ten-thousand frames per field of view (20.48 × 20.48 μm), with a maximum of 350,000 blinks recorded.

Data were processed with built-in software (softWoRx, GE Healthcare) using a Dense Stochastic Sampling Imaging (DSSI) algorithm and multiple Gaussian fits to localize events. Drift correction was performed with a model-based algorithm. Average localization precision was 21.6 nm for the events included in the final reconstructions.

Images were further processed using a custom analysis program written in Python, which was similar to one previously employing scikits-image, scipy.spatial and Opencv (http://opencv.willowgarage.com/) modules (*Macquaide et al., 2015*). These algorithms are publicly available in an online repository, (https://github.com/TerjePrivate/Ryanodine_Receptor_Dispersion_during_Heart_Failure) (*Kolstad, 2018*; copy archived at https://github.com/elifesciences-publications/Ryanodine_Receptor_Dispersion_during_Heart_Failure). Images with 10 × 10 nm pixels were convolved with a 2D Gaussian function equal to the calculated resolution of the image (~20 nm), and downscaled by a factor of 3 to produce a final pixel size of 30 × 30 nm. RyR locations were defined using a modified automated thresholding algorithm (*Kolstad, 2018*) (Otsu method), excluding the brightest 0.3% of the signal. This prevented skewing of the threshold by regions which were constantly in the active state. To minimize inclusion of autofluorescence artefacts, blinks appearing in 10 or more

consecutive frames were excluded from the final reconstruction. Acquired images were fitted to a 30 × 30 nm grid corresponding to the quatrefoil structure of the RyR protein (*Baddeley et al., 2009*). An RyR was counted as present if >half the area of a 30 nm square was above threshold, and RyR clusters were defined by occupied, neighbouring grid positions. Based on previous (*Macquaide et al., 2015*) and present (*Figure 4—figure supplement 1*) calculations clusters with edge-to-edge distance <150 nm were assumed to cooperatively generate $Ca^{2+}$ sparks, and grouped into CRUs accordingly. A stricter CRU definition, with edge-to-edge distances < 100 nm, (*Baddeley et al., 2009*; *Hou et al., 2015*) was also examined. Inter-cluster distances were calculated from the centroid of each cluster. CRU solidity was calculated as the proportion of the bounding polygon (convex hull method) which was filled with RyRs; clusters containing less than 5 RyRs were excluded from this calculation. The solidity ratio was 1 if totally filled and 0 if completely empty; therefore, lower values indicate greater CRU fragmentation. Of note, all analyses of RyR localization were performed by automated protocols in a blinded manner.

To address whether unspecific secondary antibody binding affected measurements of RyR configurations, dSTORM imaging of cardiomyocytes was performed in the absence of primary antibody. The obtained signal was then added to RyR-labeled images obtained by the standard protocol (primary plus secondary antibodies), and RyR configuration was analyzed. Non-specific labeling was observed to only negligibly increase the number of RyRs/cluster and RyRs/CRU by 3.7% and 3.2%, respectively. Similarly, RyR density was increased by 0.5%, and inter-cluster distance was reduced by 0.2%, supporting that unspecific labelling had a minute influence on the dataset.

## $Ca^{2+}$ spark and transient imaging and analysis

Using an LSM 7Live confocal microscope (Zeiss), $Ca^{2+}$ sparks were recorded from quiescent cardiomyocytes loaded with fluo-4 AM (20 µmol/L, Molecular Probes, Eugene, OR) and superfused with a HEPES Tyrode solution containing (in mmol/L): 140 NaCl, 1.0 $CaCl_2$, 0.5 $MgCl_2$, 5.0 HEPES, 5.5 glucose, 0.4 $NaH_2PO_4$, 5.4 KCl, pH 7.4, 37°C. Scans were performed with a 1024 pixel line drawn along the longitudinal axis of the cell with a temporal resolution of 1.5 ms. $Ca^{2+}$ sparks were analysed with a custom program (CaSparks 1.01, D. Ursu, 2003), as previously (*Louch et al., 2013*). Sparks were defined as local increases in fluorescence with a minimum amplitude ($\Delta F/F_0$) of 0.4, to minimise the inclusion of false positives. Linescan images of cells obtained during inhibition of $Ca^{2+}$ sparks (prolonged exposure to 10 mM caffeine) confirmed the appropriateness of this detection threshold. $Ca^{2+}$ spark frequency was normalized to cell length and recording time, and spark geometry was assessed by measurements of time to peak (TTP), full width at half maximum (FWHM), and full duration at half maximum (FDHM). Spark-mediated $Ca^{2+}$ leak was calculated as the product of spark mass (amplitude X FWHM X FDHM) and frequency.

$Ca^{2+}$ transients were elicited by field-stimulation through a pair of platinum wires (3 ms suprathreshold current pulses at 1 Hz), and recorded as confocal linescans under the same experimental conditions as $Ca^{2+}$ spark measurements. Global $Ca^{2+}$ transient characteristics were analyzed by averaging the $Ca^{2+}$ signal along the linescan, with measurements of transient magnitude (normalized to resting fluorescence, $F/F_0$), time to half maximal fluorescence ($TTF_{50}$), and TTP. Local $Ca^{2+}$ transients were averaged across narrow 2 µm bands of the linescan. Synchrony of $Ca^{2+}$ release was assessed as previously described (*Louch et al., 2006*), by plotting the profile of $TTF_{50}$ measurements across the cell and measuring the standard deviation of these values (the 'dyssynchrony index').

SR $Ca^{2+}$ content was assessed by rapidly applying 10 mM caffeine and measuring the amplitude of the elicited $Ca^{2+}$ transient.

## Microsomal $Ca^{2+}$ uptake, leak and release

$Ca^{2+}$ handling was additionally examined using crude homogenates from rat left ventricle, based on methods described by O'Brien and modified by Li *et al.* (*O'Brien, 1990*; *Li et al., 2002*). Fresh ventricular tissue was weighed and homogenized in ice cold buffer (1:10 wet weight/vol, pH 7.9) containing (in mmol/L): 300 sucrose, 5 $NaN_3$, 1 EDTA, 40 L-histidine, 40 Tris HCl and protease inhibitors. Homogenization was performed with a Polytron 1200 (Kinematica AG, Luzern, Switzerland) at 25000 rpm for 3 × 20 s, with a 20 s break between bursts. Homogenates were then aliquoted, frozen in liquid $N_2$, and stored at −80°C until use.

Ca$^{2+}$ uptake and release were measured in 2.2 ml of assay buffer, containing (in mmol/L): 165 KCl, 22 Hepes, 7.5 oxalate, 11 NaN$_3$, 0.0055 TPEN, 4.5 MgCl$_2$, 9 Tris HCl and 0.002 fura-2 salt (pH = 7.0, 37°C). Ca$^{2+}$ fluxes were monitored with an LS50B luminescence spectrometer (Perkin Elmer Ltd, Beaconsfield, Buckinghamshire, United Kingdom) after addition of 100 µl of freshly-thawed and vortexed homogenate. Ca$^{2+}$ uptake by the vesicles was initiated by addition of Na$_4$ATP (2.2 mmol/L), and then blocked by application of thapsigargin (1.5 µmol/L) to assess RyR leak. Releasable SR Ca$^{2+}$ content was estimated by measuring Ca$^{2+}$ release induced by application of the RyR opener 4-chloro-m-cresol (4-CMC) (5.5 mmol/L). The fluorescence ratio was calibrated to [Ca$^{2+}$] using the following equation: [Ca$^{2+}$]=K$_d$ *((R - R$_{min}$)/(R$_{max}$ - R))*(S$_{f2}$/S$_{b2}$), where R is the 340 nm/380 nm fluorescence ratio, K$_d$ is the dissociation constant of fura-2 and S$_{f2}$/S$_{b2}$ is the ratio of measured fluorescence intensity at 380 nm when fura-2 is Ca$^{2+}$ free or saturated, respectively. R$_{min}$ is the ratio at very low [Ca$^{2+}$]$_i$ and R$_{max}$ is the ratio at saturating [Ca$^{2+}$]$_i$, obtained by adding 3.3 mmol/L EGTA and 4.8 mmol/L CaCl$_2$ respectively to the cuvette at the end of each recording.

## Western blotting

Frozen tissue from rat left ventricles was homogenized in cold buffer (210 mM sucrose, 2 mM EGTA, 40 mM NaCl, 30 mM HEPES, 5 mM EDTA) with the addition of a Complete EDTA free protease inhibitor cocktail tablet (Roche Diagnostics, Oslo, Norway) and a PhosSTOP tablet (Roche). SDS was then added to the homogenates to a final concentration of 1%, and protein concentrations were quantified using a micro BCA protein assay kit (Thermo Fischer Scientific Inc, Rockford, IL). Bovine serum albumin (BSA) was used as standard protein.

The following primary antibodies were employed for immunoblotting: RyR (1:1000; MA3-916, Thermo Scientific), BIN1 (1:500; sc23918, Santa Cruz Biotechnology), junctophilin-2 (1:1000; sc-51313, Santa Cruz Biotechnology) and GAPDH (1:500; sc-20357, Santa Cruz Biotechnology). Secondary antibodies were anti-rabbit (NA934V, GE Healthcare), anti-mouse (NA931V, GE Healthcare) or anti-goat (HAF109, R and D Systems) IgG-HRP linked whole antibody. Data were normalized to GAPDH and then to Sham values (*Figure 2—figure supplement 1*).

Protein homogenates (5 or 15 µg/lane) were size fractionated on 4–15% or 15% Criterion TGX gels (Biorad Laboratories, Oslo, Norway) and transferred to 0.45 µM PVDF-membranes (GE Healthcare). The membranes were blocked in 5% non-fat milk or 5% Casein (Roche Diagnostics) in Tris-buffered saline with 0.1% Tween (TBS-T) for 1 hr at room temperature, and then incubated with primary antibody overnight at 4°C. Secondary antibodies were incubated for 1 hr at room temperature and blots were developed using Enhanced Chemiluminescence (ECL prime, GE healthcare). Chemiluminiscense signals were detected by a LAS 4000 (GE healthcare) and protein levels were quantified using ImageQuant software (GE Healthcare).

## Mathematical model

A mathematical model was created to simulate the effects of varied RyR localization and CRU geometry on Ca$^{2+}$ spark characteristics. We have made all simulation results, geometries, and code specific to this study available in an online repository (*Kolstad, 2018*), along with code for the full reaction-diffusion simulator. The model extended from the work of *Hake et al. (2012)*, with included RyR stochasticity developed from previous work by Cannell and colleagues (*Cannell et al., 2013*). We have chosen this model for the relative simplicity of its gating (no direct inter-RyR coupling, or explicit luminal Ca$^{2+}$ regulation), and because it was built for a similarly constructed (spatially discretized) geometry, (*Cannell et al., 2013*) unlike most other recent RyR2 gating (*Williams et al., 2011*; *Wescott et al., 2016*). A set of coupled partial differential equations was employed to describe the temporal evolution of the free and bound [Ca$^{2+}$] in the SR and cytosol:

$$\frac{\partial c}{\partial t} = D_c \nabla^2 c - \sum_{i=1}^{4} R_i(c, b_i), x \in \Omega_C$$

$$\frac{\partial b_i}{\partial t} = D_i \nabla^2 b_i + R_i(c, b_i), i = 1, 2, 3, 4 x \in \Omega_C$$

$$\frac{\partial s}{\partial t} = D_s \nabla^2 s - R_5(s, b_5), x \in \Omega_S$$

$$\frac{\partial b_5}{\partial t} = R_5(s, b_5), x \in \Omega_S$$

Here $\Omega_C$ is this cytosolic domain, including the cleft space, and $\Omega_S$ is the SR, including both junctional and network SR components. Four buffers were included in $\Omega_C$: ATP, calmodulin, troponin and Fluo-4, and one buffer, calsequestrin, was included in $\Omega_S$. These buffers are numbered from 1 to 5 and their corresponding concentrations are denoted $b_i$. Troponin and calsequestrin were regarded to be stationary, and the corresponding diffusion coefficients ($\sigma$) were therefore set to zero (see **Supplementary file 1**). The calcium concentrations in $\Omega_C$ and $\Omega_S$ are denoted $c$ and $s$ respectively.

The buffering reactions are of the form

$$R(c, b_i) = k_{on} c (B_{tot} - b_i) - k_{off} b_i$$

where $B_{tot}$ is the total buffer concentration, and $k_{on}$ and $k_{off}$ are the on and off rates for the buffer, respectively.

The two domains are coupled through a flux condition over the SR membrane:

$$D_c \frac{\partial c}{\partial n} = -D_s \frac{\partial s}{\partial n} = J(c, s)$$

where

$$J(c, s) = \begin{cases} J_{RyR} & x \in \Gamma_{RyR} \\ J_{Serca} & x \in \Gamma_{Serca} \\ 0 & elsewhere \end{cases}$$

The RyR flux is computed by:

$$J_{RyR}(c, s) = g_{RyR} \gamma (c - s)$$

where $\gamma \in 0, 1$ is a stochastic variable that switches between the conductive (O) and non-conductive (C) states according to:

$$C \underset{k^-}{\overset{k^+}{\rightleftarrows}} O.$$

While we have chosen a different form for the equations expressing the default transition rates, they are equivalent to those in the original model of **Cannell et al. (2013)** with the exception that we have set limits to both $k^+$ and $k^-$ at low dyadic calcium:

$$k^+(c) = f\left(\left(\frac{c}{K^+}\right)^{n^+}, k_{min}^+, k_{max}^+\right)$$

$$k^-(c) = f\left(\left(\frac{c}{K^-}\right)^{n^-}, k_{min}^-, k_{max}^-\right)$$

where for $a<b$:

$$f(y, a, b) = \begin{cases} a, & if & y < a \\ y, & if & a \leq y \leq b \\ b, & if & y \end{cases}$$

The parameters are given in **Supplementary file 1**.
The SERCA formulation is taken from **Tran et al. (2009)** and is of the form:

$$J_{Serca}(c,s) = \frac{a_1 c^2 - a_2 s^2}{a_3 c^2 + a_4 s^2 + a_5}.$$

## Geometry, numerics, and implementation

The computational domain in our model ($\Omega_S \cup \Omega_c$) was a (2 µm) cube containing a single CRU (Illustrated in *Figure 4—figure supplement 1A*). Unlike the original work of Cannell and colleagues, for which RyR locations were fixed for all simulations, our simulations involve algorithm-defined changes in the jSR geometry and location of RyRs to reflect the structural differences captured by the dSTORM recordings. In all geometries the domain consisted of a 12 nm wide cleft space sandwiched between the junctional SR (jSR) surface and the t-tubular surface. The latter was represented as a non-conductive slab inside $\Omega_c$, serving as a barrier to diffusion. RyRs were located on the opposing jSR surface, each occupying a space 36 × 36 nm, with neighbouring RyRs placed 36 nm apart (centre-to-centre distance). The jSR was modeled as a physical extension ('padding') around each RyR by a defined distance equivalent to 1 RyR diameter (36 nm). As mentioned above, RyRs were arranged according to dSTORM-derived locations, and the jSR shape was adjusted according to the RyR locations. In some simulations (*Figure 4A*, *Figure 6—source data 1*) the geometries of the jSR were fixed while idealized RyR lattice geometries were varied to explore the independent effect of modifying RyR number with fixed jSR volume and the locally releasable calcium pool (*Figure 4A*). To specifically investigate the effects of RyR dispersal, example dSTORM-identified RyR patterns were selected with similar total RyR number but with arrangement into a varied number of sub-clusters (1, 3, 7, or 10). The ratio of RyR number to jSR volume for these example CRUs is presented in *Figure 4—source data 1*. To limit the effect variation in jSR volume has on the releaseable Ca$^{2+}$ store, we fixed the total SR volume (and thus initial SR Ca$^{2+}$ content) across all geometries by modifying the non-junctional SR (nSR) volume as required. Of note, while nSR concentration was modeled as a continuum, global SR calcium concentration was effectively clamped at initial values as expected for the time-scale and spatial-scale simulated.

Dyadic Ca$^{2+}$ release was initiated by opening a single RyR in the CRU; this 'trigger' RyR was selected randomly and varied between consecutive simulations. Simulated triggered Ca$^{2+}$ release from the CRU was then allowed to proceed, with the above equations discretized in space with a finite volume approach (12 nm edge length throughout the domain), and solved in time using explicit Euler time stepping. Specifically, we used operator splitting and solved each of the reaction and diffusion sub-problems with a fixed $\Delta t = 0.1$µs, except when calculating the RyR release current. Due to the small element volumes and high fluxes this calculation is very stiff, so instead we solve it analytically. If we use x to denote the SR calcium concentration and y to denote the cleft calcium concentration, we can write this sub-problem as:

$$\dot{x} = K(y - x)$$

$$\dot{y} = K(x - y)$$

where K is the channel conductance per element volume. The solution to this subsystem is given by:

$$x(t) = S - De^{-2Kt}$$

$$y(t) = S + De^{-2Kt}$$

Where:

$$S = \frac{(y(0) + x(0))}{2}$$

$$D = \frac{(y(0) - x(0))}{2}$$

Using this scheme it is possible to take arbitrarily long time steps without introducing instabilities.

For the RyR gating model half-maximal activation was achieved at 80 µM $Ca^{2+}$, which allowed cooperative opening of adjacent RyR clusters located up to ≈150 nm apart (ie. 4 RyR lengths) if the clusters shared jSR (*Figure 4—figure supplement 1B,C*). Thus, we defined CRUs as groupings of RyR clusters with edge-to-edge distances < 150 nm, in agreement with recent work (*Macquaide et al., 2015*), but also compared data with a stricter CRU definition (cluster distances < 100 nm) employed in other publications (*Baddeley et al., 2009*; *Hou et al., 2015*). RyR rates are shown in *Supplementary file 2*.

## Statistical analyses

All results are expressed as mean values ± standard error of the mean. All statistical significance was calculated in SigmaPlot by Student's t-test or ANOVA with Bonferroni post-hoc comparison for normally distributed data, as appropriate. Skewed distributions of experimental and modelled $Ca^{2+}$ spark parameters were respectively assessed by the nonparametric Mann-Whitney Rank Sum Test and Kruskal-Wallis ANOVA with Dunn's test for post-hoc comparisons. dSTORM-based measurements of RyR geometries were compared with averages taken both across cells and animals, with respective statistical testing by t-tests and linear mixed effects models (*Lindstrom and Bates, 1988*). Statistical significance was defined as $p < 0.05$.

## Access to raw experimental data and analysis source code

All raw data acquired and analyzed in this study are publicly available at https://github.com/TerjePrivate/Ryanodine_Receptor_Dispersion_during_Heart_Failure (*Kolstad, 2018*; copy archived at https://github.com/elifesciences-publications/Ryanodine_Receptor_Dispersion_during_Heart_Failure).

## Acknowledgements

The authors thank the Section of Comparative Medicine, Oslo University Hospital Ullevål for animal care and Dina Behmen and Almira Hasic at the Institute for Experimental Medical Research, Oslo University Hospital Ullevål for assistance with Western blotting. We additionally thank Vigdis Sørensen at the Advanced Light Microscopy Core Facility, Oslo University Hospital, Montebello, for assistance with dSTORM imaging.

## Additional information

### Funding

| Funder | Grant reference number | Author |
|---|---|---|
| Horizon 2020 Framework Programme | Consolidator grant for WEL 647714 | Terje R Kolstad<br>William E Louch |
| Helse Sør-Øst RHF | 2012072 | Terje R Kolstad<br>Ole M Sejersted<br>William E Louch |
| Helse Sør-Øst RHF | 2013014 | Terje R Kolstad<br>Ole M Sejersted<br>William E Louch |
| Oslo University Hospital Ullevål | | Terje R Kolstad<br>Ole M Sejersted<br>William E Louch |
| University of Oslo | | William E Louch<br>Terje R Kolstad |

The funders had no role in study design, data collection and interpretation, or the decision to submit the work for publication.

### Author contributions

Terje R Kolstad, Conceptualization, Data curation, Formal analysis, Validation, Investigation, Visualization, Methodology, Writing—original draft, Writing—review and editing; Jonas van den

Brink, Conceptualization, Software, Formal analysis, Investigation, Visualization, Methodology, Writing—original draft, Writing—review and editing; Niall MacQuaide, Conceptualization, Software, Formal analysis, Investigation, Visualization, Writing—original draft, Writing—review and editing; Per Kristian Lunde, Formal analysis, Investigation, Methodology, Writing—original draft, Writing—review and editing; Michael Frisk, Investigation, Visualization, Writing—original draft, Writing—review and editing; Jan Magnus Aronsen, Investigation, Methodology, Writing—original draft, Writing—review and editing; Einar S Norden, Formal analysis, Methodology, Writing—original draft, Writing—review and editing; Alessandro Cataliotti, Ole M Sejersted, Resources, Supervision, Writing—original draft, Writing—review and editing; Ivar Sjaastad, Conceptualization, Resources, Supervision, Methodology, Writing—original draft, Writing—review and editing; Andrew G Edwards, Conceptualization, Formal analysis, Investigation, Visualization, Methodology, Writing—original draft, Writing—review and editing; Glenn Terje Lines, Conceptualization, Software, Formal analysis, Investigation, Methodology, Writing—original draft, Writing—review and editing; William E Louch, Conceptualization, Resources, Data curation, Formal analysis, Supervision, Funding acquisition, Validation, Investigation, Visualization, Methodology, Writing—original draft, Project administration, Writing—review and editing

### Author ORCIDs
Terje R Kolstad http://orcid.org/0000-0002-0589-5689
Ole M Sejersted https://orcid.org/0000-0001-8817-3296
William E Louch http://orcid.org/0000-0002-0511-6112

### Ethics
Animal experimentation: All experiments were approved by the Norwegian National Animal Research Authority (project license no. FOTS 5982, 7786), and were performed in accordance with the National Institute of Health guidelines (NIH publication No. 85-23, revised 2011) and European Directive 2010/63/EU.

### Decision letter and Author response
Decision letter https://doi.org/10.7554/eLife.39427.021
Author response https://doi.org/10.7554/eLife.39427.022

## Additional files

### Supplementary files
• Supplementary file 1. Buffer parameters.
DOI: https://doi.org/10.7554/eLife.39427.017

• Supplementary file 2. RyR rates.
DOI: https://doi.org/10.7554/eLife.39427.018

• Transparent reporting form
DOI: https://doi.org/10.7554/eLife.39427.019

### Data availability
Source data files have been provided for Figures 2, 4 and 6. All raw data acquired and analyzed in this study are publicly available in the following repository: https://github.com/TerjePrivate/Ryano-dine_Receptor_Dispersion_during_Heart_Failure (copy archived at https://github.com/elifesciences-publications/Ryanodine_Receptor_Dispersion_during_Heart_Failure).

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
