## [Decision Letter]

Thank you for submitting your article "Ryanodine Receptor Dispersion Disrupts Ca^2+^ Release in Failing Cardiac Myocytes" for consideration by *eLife*. Your article has been reviewed by Harry Dietz as the Senior Editor, a Reviewing Editor, and three reviewers. The following individual involved in review of your submission has agreed to reveal his identity: David Baddeley (Reviewer #3).

The reviewers have discussed the reviews with one another and the Reviewing Editor has drafted this decision to help you prepare a revised submission.

Summary:

This is an interesting and thoughtful manuscript that advances our understanding of the changes underlying the slower and weaker calcium transients in failing cardiac myocytes. The authors describe changes in the arrangement of ryanodine receptors, using super-resolution microscopy and simulations. The combination of structural cluster imaging, functional Ca^2+^ imaging and simulations has been used effectively, and the results are convincing. The main finding of the study is that RyR clusters are more dispersed in the failing than in normal cardiomyocytes.

Essential revisions:

1) Cluster fragmentation data (Figure 2) need to be shown more clearly and in greater detail. Some of the magnified unthresholded data and the scaled raw image data should be displayed (taking care not to saturate too much). A clear case should be made that the fragmentation is not a thresholding artifact, as it is known that threshold changes could cause apparent fragmentation for a given data set.

2) It is critical that the authors provide enough implementation detail for the simulation model so that a reader can use the simulation for stably dealing with the nonlinearities contained in a stochastic RyR switching model. If a custom code was used, it should be provided.

It should be clarified if the basic geometry is the same as in the Laver et al. 'induction decay' model, or if not – what is the scope and rationale for the changes made. Also, the authors should explain how they performed the numerical integration (was the time step fixed or adaptive; how were the sudden changes in transition rates incorporated in to the discretized solver?) The authors should also provide rationale for the data exclusions and clearly specify which data were used.

3) The method is based on the assumption that peripheral couplons are effectively 2D objects parallel to the cell surface and hence the imaging plane. It becomes increasingly hard to justify this assumption when moving away from surface where EM data shows that couplons take on complex 3D geometries, often wrapping around t-tubules. The absence of double-rows would imply that the images taken in this paper are not at the cell surface. While looking at the projected area will still give a reasonable estimate of relative numbers, it is likely that the absolute number of RyRs is underestimated for internal couplons. Regarding the projections to a regular 30nm grid, there is very limited evidence that RyRs form a regular grid in normal conditions within cardiomyocytes. Even if they do form a regular grid, it likely does not align with the imposed sub-sampled pixel grid. The authors should specifically discuss these important aspects of their model. It is also suggested to acknowledge that the estimated numbers are not necessarily very precise and discuss the effect that changing the RyR number would have on the simulation results.

4) With regard to the measured parameters, simple t-tests do not suffice. A hierarchical nested model should be used as p<0.05 is not satisfactory when there is large inter-group variability and the number of animals is small. The data needs to be reanalyzed, e.g. using methods described in a recent publication (Sikkel et al., 2017).

---

## [Author Response]

Essential revisions:1) Cluster fragmentation data (Figure 2) need to be shown more clearly and in greater detail. Some of the magnified unthresholded data and the scaled raw image data should be displayed (taking care not to saturate too much). A clear case should be made that the fragmentation is not a thresholding artifact, as it is known that threshold changes could cause apparent fragmentation for a given data set.

Thank you for your helpful comments, which have improved the manuscript. Figure 2 has now been changed to include a new panel which displays a high magnification “high-low/grayscale” image of the region of interest. This has been done to show that the binarized images give a reasonable estimation of RyR clusters based on the raw images for both Sham and HF groups. The added panel has been given a “high-low” look-up table wherein the top and bottom 3% grey value pixels are colored red and blue respectively. This more clearly illustrates that the grayscale images from which the binarization was derived are not heavily saturated.

2) It is critical that the authors provide enough implementation detail for the simulation model so that a reader can use the simulation for stably dealing with the nonlinearities contained in a stochastic RyR switching model. If a custom code was used, it should be provided.It should be clarified if the basic geometry is the same as in the Laver et al. 'induction decay' model, or if not – what is the scope and rationale for the changes made. Also, the authors should explain how they performed the numerical integration (was the time step fixed or adaptive; how were the sudden changes in transition rates incorporated in to the discretized solver?) The authors should also provide rationale for the data exclusions and clearly specify which data were used.

We have now elaborated on each of these points in the Materials and methods section of the manuscript. For the reviewers' reference, the time step was fixed at 0.1 microseconds, which is sufficient for the RyR gating events themselves, and is not limiting for solution of the resulting concentration changes on either side of the jSR membrane, because this sub-problem can be solved analytically.

3) The method is based on the assumption that peripheral couplons are effectively 2D objects parallel to the cell surface and hence the imaging plane. It becomes increasingly hard to justify this assumption when moving away from surface where EM data shows that couplons take on complex 3D geometries, often wrapping around t-tubules. The absence of double-rows would imply that the images taken in this paper are not at the cell surface. While looking at the projected area will still give a reasonable estimate of relative numbers, it is likely that the absolute number of RyRs is underestimated for internal couplons. Regarding the projections to a regular 30nm grid, there is very limited evidence that RyRs form a regular grid in normal conditions within cardiomyocytes. Even if they do form a regular grid, it likely does not align with the imposed sub-sampled pixel grid. The authors should specifically discuss these important aspects of their model. It is also suggested to acknowledge that the estimated numbers are not necessarily very precise and discuss the effect that changing the RyR number would have on the simulation results.

The reviewers are correct that the RyR clusters imaged in our work are near the surface, but not at the surface. We agree that using grid-based counting methods for determining RyR numbers at internal locations is likely more error prone and may underestimate cluster sizes. The following section has been added to the Discussion section to address these points:

“However, it should be noted that the grid-based method for RyR counting assumes that the channels lie parallel to the field of view, with a uniform, grid-like configuration; presumptions which are likely less valid when RyRs are visualized internally than at the cell surface. Due to super-imposition of internal RyRs along the z axis, it is therefore likely that present RyR cluster sizes are somewhat underestimated.”

We have made a similar statement in the Discussion section regarding estimates of RyRs / CRU:

“Despite possible underestimation of RyR numbers / CRU due to methodological issues noted above, these estimates are in relatively close agreement with an electron microscopy tomography study of mouse ventricular cardiomyocytes (Hayashi et al., 2009)”.

We have additionally considered the reviewer’s point that underestimation of RyR cluster and CRU sizes by grid-based counting procedures may influence the simulation results. To this end, we have performed additional modeling where RyR number was varied, while maintaining the ratio of jSR to nSR. These new data (Figure 6—source data 1) showed that for CRUs larger than 20 RyRs there was little effect of varying RyR number on Ca^2+^ spark kinetics. Mean CRU sizes observed experimentally in both Sham and HF cardiomyocytes were within this range (indicated as dotted vertical lines in Figure 6—source data 1). Furthermore, since RyR numbers were likely *underestimated* experimentally, we do not expect that such counting errors have marked consequences regarding the effects of CRU dispersal on Ca^2+^ spark parameters. Of note, a lack of sensitivity of spark characteristics to RyR number in average-sized clusters is in agreement with previous work (Cannell et al., 2013). This finding has been described in the Results section:

“Of note, although CRUs were observed to contain fewer RyRs in HF than Sham (Figure 2C), simply reducing the RyR number to an equivalent degree in the mathematical model did not markedly alter Ca^2+^ spark kinetics (Figure 6—source data 1), further confirming a key role of CRU fragmentation in failing cells.”

We have also addressed this point in the Discussion section:

“Our modeling results also suggest that an altered number of RyRs / CRU, due either to methodological under-estimation or loss of RyRs from CRUs in HF, will not in and of itself promote slowing of sparks kinetics (Figure 6—source data 1). Indeed, we observed that spark kinetics were relatively insensitive to changes in RyR number for medium-sized CRUs, in agreement with previous work (Cannell et al., 2013).”

4) With regard to the measured parameters, simple t-tests do not suffice. A hierarchical nested model should be used as p<0.05 is not satisfactory when there is large inter-group variability and the number of animals is small. The data needs to be reanalyzed, e.g. using methods described in a recent publication (Sikkel et al., 2017).

A hierarchical “nested ANOVA” linear mixed-effects model has been applied to the dataset, which allows RyR configuration to be cumulated and compared within animals. The resulting p-values can be found in a table included in Figure 2—source data 1 and are presented alongside data cumulated within cells. The main differences in RyR configuration between Sham and HF were similar based on these two approaches.